# Assessment of current good manufacturing practice (cGMP) compliance in pharmaceutical manufacturers in Ethiopia: Cross-sectional descriptive study

Teka Benti Adola [1,2*], Desta Assefa[1], Fikadu Ejeta[1], Fanta Gashe[1], Getahun Paulos[1,3], Dereje Kebebe Borga[1,4*]

1 School of Pharmacy, Faculty of Health Sciences, Institute of Health, Jimma University, Jimma, Ethiopia, 2 School of Pharmacy, College of Health Sciences, Wollega University, Nekemte, Ethiopia, 3 Department of Pharmacy, Komar University of Science and Technology, Sulaymaniyah, Iraq, 4 Department of Pharmacy, College of Applied Natural Sciences, Adama Science and Technology University, Adama, Ethiopia

* Teka.B2025@wollegauniversity.edu.et (TB); dereje.keborg@gmail.com (DKB)

## Abstract

### Background

The distribution of low-quality medications poses a serious risk to public health, especially in underdeveloped nations like Ethiopia. Ineffective use of Current Good Manufacturing Practices (cGMP) increases these dangers, which include cross-contamination, mix-ups, and incorrect labeling. To ensure consistent product quality and adherence to set standards, a Pharmaceutical Quality System (PQS) that effectively integrate cGMP is essential. Assessing cGMP compliance is essential for the reliable production of safe pharmaceuticals.

### Objective

This study aims to assess cGMP compliance in Ethiopian pharmaceutical manufacturers.

### Methods

A cross-sectional descriptive design was employed for this study. The data collection process involved a checklist for field observations and, interview-based questions. The checklist, developed based on WHO GMP requirements, was filled through direct site observations, document reviews, and discussions with key persons in the appropriate departments. The study included six companies located in Addis Ababa and its surrounding areas. The qualitative data were transformed into quantitative data through coding and categorization for easier analysis. The data were then analyzed using descriptive statistics, cross-tabulation, and other statistical tests with

**Data availability statement:** All relevant data are within the manuscript and its Supporting Information files.

**Funding:** The author(s) received no specific funding for this work.

**Competing interests:** The authors have declared that no competing interests exist.

**Abbreviations:** cGMP, current good manufacturing practice; EFDA, ethiopian food and drug administration; FDA, food and drug administration; GMP, good manufacturing practice; HVAC, heating, ventilation, and air conditioning; PQS, pharmaceutical quality system; QA, quality assurance; QRM, quality risk management; WHO, World Health Organization.

SPSS software version 28. The results were displayed using data visualization tools like tables and graphs.

## Results

The study revealed overall cGMP compliance rates of 62.8%, 85.84%, 84.95%, 86.72%, 84.95%, and 84.1% for companies 1–6, respectively. All companies had some degree of GMP non-conformity; however, only Company 1 had critical deviations. The GMP standards were well-maintained by the remaining five companies. Company 1 has to make improvements in several areas, most notably its QA systems. Additionally, Companies 1, 5, and 6 must improve their QRM systems. Company 5 needs to improve its production process, while Companies 1, 2, and 3 should focus on better managing their equipment and materials.

## Conclusion and recommendation

This study evaluated GMP compliance levels across six pharmaceutical manufacturers in Ethiopia. Most of the assessed companies showed a satisfactory degree of GMP compliance. To improve compliance, minimize risks, and ensure product quality, safety, and operational efficiency, the study suggested strengthening risk mitigation measures such as raising employees' knowledge and training.

## Introduction

The pharmaceutical manufacturer is a firm that produces and markets medications, chemicals, and medical supplies for use in disease prevention, diagnosis, and treatment. This pharmaceutical company requires guidelines and regulatory oversight to ensure the production of quality-standard products [1].

A part of the quality management system (QMS), the PQS is a comprehensive framework that ensures the safety and quality of pharmaceutical products. According to the PQS Concepts, the manufacturer is responsible for pharmaceutical product quality, ensuring that it is safe for intended use, complying with all marketing authorization requirements, and preventing patient risks from insufficient safety or quality. Senior management is tasked with attaining this quality standard, requiring active involvement and dedication from employees across all departments and organizational levels, as well as from suppliers and distributors. A well-planned and executed PQS that integrates GMP and QRM is essential to consistently meet intended quality [2,3].

The quality assurance method, known as current Good Manufacturing Practices, or cGMP, ensures that pharmaceuticals are consistently produced and controlled to fulfill their intended usage, product specifications, marketing authorization, or clinical trial authorization. In addition to defining, validating, reviewing, and documenting the processes necessary for manufacturing and testing, cGMP includes quality measurements for both quality control and production. It also ensures that personnel,

buildings, and materials are suitable for the production of pharmaceutical products. Legal provisions covering distribution duties, contract production, testing, and management of product defects and complaints are also included in cGMP. To ensure product efficacy, safety, and quality, cGMP primarily regulates and minimizes the risks involved in pharmaceutical manufacturing [1,3].

cGMP mandates rigorous testing of both raw materials and finished products to ensure their quality, potency, purity, and safety. Throughout the manufacturing process, every product must adhere to a precise master formula without any deviations. This necessitates a robust quality assurance system that undergoes frequent testing, ongoing evaluations, and precise compliance at every production stage [1]. Manufacturers are required to meticulously follow and document each manufacturing step. Documentation plays a crucial role in compliance, with auditors conducting regular inspections to ensure quality and consistency in laboratories and other facilities [4].

cGMP's main components include personnel, products, premises, processes, and procedures. The company needs to have an adequate number of qualified individuals with the right experience. Ensuring the quality of pharmaceutical products relies on having the correct personnel for manufacturing and quality control. The manufacturer is responsible for training all staff in production areas and quality control laboratories, including technical, maintenance, cleaning personnel, and other necessary staff. Continuous training is essential, ensuring that employees improve their job performance through ongoing learning assessments [3].

Manufacturers can further enhance this by providing all team members access to a written outline of the complete manufacturing process. Essentially, processes must adhere to up-to-date procedures, with personnel possessing the necessary knowledge of them. A standard operating procedure comprises essential instructions for executing a process to achieve a specific outcome. Maintaining the condition of buildings, labs, and equipment is crucial for safe and efficient manufacturing. Proper maintenance, cleanliness, and timely repairs not only reduce the risk of equipment failures but also ensure consistent outcomes. The primary objective is to minimize product variations while simultaneously protecting customers, staff, and patients from operational challenges within the facility [3,4].

Historically, most Ethiopian pharmaceutical manufacturers have produced drugs that do not meet GMP standards, which has led to the distribution of low quality of products. Patients who use these subpar products may experience several adverse effects, many of which are not immediately apparent. Patients rely on health professionals for advice as they are unable to identify quality issues with medications based on characteristics such as color or scent. In turn, healthcare workers place their trust in pharmaceutical manufacturers who are responsible for determining and maintaining the quality of pharmaceuticals during the manufacturing process [5].

When pharmaceuticals are not manufactured in compliance with GMP guidelines, it can lead to the production of defective products that may not be safe or effective. This, in turn, can result in antimicrobial resistance, subpar treatment outcomes, and unfavorable drug interactions. Additionally, by raising the expense of therapy, noncompliance indirectly harms the general population. A company's reputation may suffer as a result of waste, product recalls, and consumer complaints brought on by noncompliance with cGMP [6]. Risks are inevitable in pharmaceutical enterprises, and diligent management and monitoring are required to keep risk at acceptable levels. it is essential to assess pharmaceutical manufacturers' cGMP compliance systems to identify areas for improvement and weaknesses [7]. There are 12 (twelve) large-scale pharmaceutical manufacturers in Ethiopia, most of which are located in Addis Ababa and its surrounding areas [8,9]. The objective of the study was to evaluate the implementation of cGMP in Ethiopian pharmaceutical manufacturers.

Understanding cGMP compliance in Ethiopia is critical because it serves as the bridge between the country's ambitious public health goals and its industrial economic strategy. To reduce Ethiopia's 85% reliance on imported drugs and break the "glass ceiling" that prevents international exports, the Ethiopian Food and Drug Authority (EFDA) enforces cGMP as a non-negotiable legal framework. Established under Proclamations No. 1112/2019 and 1263/2021, the EFDA serves as the nation's autonomous gatekeeper, utilizing a comprehensive toolkit of on-site inspections, market authorizations, and corrective and preventive action plans to ensure all pharmaceuticals meet rigorous safety standards [10]. Achieving these

WHO-level benchmarks is the only pathway for local firms to secure WHO Prequalification, compete in global markets, and mitigate the public health risks of substandard medicines or antimicrobial resistance. To maintain this integrity, the EFDA enforces a strict penalty system where non-compliance results in immediate administrative actions—such as the revocation of certificates and product recalls—alongside severe criminal sanctions, including heavy fines and imprisonment, ultimately transforming Ethiopia's botanical and scientific heritage into a resilient, self-sufficient healthcare supply chain [10,11].

## Methods

A cross-sectional descriptive design was employed to study large-scale pharmaceutical manufacturers in Ethiopia involved in the production and marketing of high-volume pharmaceuticals in various dosage forms. The study included manufacturers that produce pharmaceuticals and excluded those that exclusively produce medical supplies.

### Sampling and participants

From an initial pool of ten companies in and around Addis Ababa, six pharmaceutical manufacturers were purposively selected. This sample size was primarily determined by practical considerations. The manufacturers were chosen based on their scale of operations, specifically their capacity to produce a high number of product batches and a wide range of dosage forms.

From each selected company, five key technical and managerial personnel directly involved in implementing product quality assurance systems were purposively chosen for interviews. These respondents were selected based on their specific roles and responsibilities within their departments.

### Variables and instruments

GMP compliance served as the dependent variable. The independent variables included GMP certificates, training, workload, Process Analytical Technology (PAT), production line suitability, government support, education level, and work experience. The study utilized two data collection instruments: a standardized checklist, developed from WHO GMP standards (due to their alignment with national guidelines) to evaluate cGMP application, and interview-based questions to gather detailed information on specific variables.

The questionnaires were specifically developed for this study, drawing upon relevant literature and guidelines, including WHO recommendations, the Pharmaceutical Inspection Convention/Cooperation Scheme, and Ethiopia's Good Manufacturing Practice for Medicinal Products.

### Data collection

Data were collected by two trained pharmacists with GMP experience using a combination of techniques at the manufacturing facilities: direct observation, document review, and conversations with key personnel. The checklist assessment and observation process rated GMP compliance by evaluating various aspects such as facility design, equipment validation, process control, and documentation practices, ensuring all activities and manufacturing processes were assessed according to GMP standards.

Once the data was collected, it was coded. The rating of the compliance status of GMP guideline was done following the steps below:

Step 1: Assessment was made as per the developed checklist.

Step 2: The compliance status of each observation was categorized as Fully compliance (FC), Partially compliance (PC), Not compliance (NC).

Step 3: Based on the compliance status of each requirement, evaluating cGMP elements were numerically rated to show the degree of compliance of each GMP element in the pharmaceutical manufacturers as follows: FC: 3, PC: 2 and, NC: 1

Finally, the total rating was computed with Equation 1 below:

$$\frac{\sum(GMP\ elements\ rated\ score)}{maximum\ possible\ score} \times 100\%$$

GMP elements rated score: The rating given to important GMP elements is based on the rules and regulations set forth by regulatory bodies.

Maximum possible score: The activity's value for full compliance with regulations.

Step 4: Observed deficiencies were categorized as critical, major, or minor based on their severity, following the evaluation of the compliance status as Fully Compliant (FC), Partially Compliant (PC), or Non-Compliant (NC) [12].

To describe the main features of the data, the gathered information was examined using suitable statistical techniques, such as descriptive statistics, which include frequencies and percentages. The qualitative data were transformed into quantitative data through coding, categorization, and the use of Likert scales for easier analysis. Crosstabulation analysis (Fisher-Freeman-Halton Exact Test) were employed to investigate the associations between various variables. The data findings were then presented clearly using tables and graphs, while the characteristics of the variables, as well as any observed patterns or trends, were described. This study was conducted from July 10 to September 16, 2024.

### Ethical considerations and consent

A letter of permission and ethical approval, bearing the reference number 'JUIH/IRB/198/24', was obtained from the Institutional Review Board (IRB) of Jimma University, Institute of Health, before data collection on May 6, 2024. The purpose of the study was thoroughly explained to the participants, and written informed consent was obtained from all study subjects. Additionally, the confidentiality and anonymity of the data were maintained by the investigators and research assistants throughout the study.

### Results

#### Profile of selected pharmaceutical manufacturing companies

The study involved a total of six manufacturing firms licensed to develop, formulate, and produce pharmaceuticals in various dosage forms. Of the six participating companies, four are GMP certified. To uphold ethical standards, company names were replaced with specific codes ranging from 1 to 6. Each company was assessed based on WHO GMP requirements, which align with the GMP standards adopted by the EFDA. The key quality standard elements were rated as "Not Compliant," "Partially Compliant," and "Fully Compliant," as presented in Table 1.

The levels of deviation that pharmaceutical manufacturers have from GMP guidelines are depicted in Fig 1. Most of the evaluated companies demonstrated a good level of compliance. However, these companies exhibited varying degrees of non-compliance, ranging from minor to critical deficiencies. While all companies showed some level of GMP deficiencies, only Company 1 exhibited critical deficiencies. The remaining five companies maintained relatively strong compliance with GMP standards.

#### Overall assessment of cGMP compliance level

The evaluation of cGMP compliance systems in the manufacturing companies was conducted using field observation checklists. The overall status of cGMP compliance is summarized in Fig 2. Each GMP sub-element of the quality system was assessed, with activities implemented under these sub-elements being evaluated for compliance. The overall compliance score for each company was assigned based on these evaluations. Company 4 stands out with the highest percentage of "Fully Compliant" (86.72%) and the lowest percentage of "Not Compliant" (0.88%). Companies 2, 3, and 5 show similar compliance profiles, characterized by a high degree of "Fully Compliant," a moderate degree of "Partially

**Table 1. General Overview of the GMP Sub-Elements Compliance Scores of Pharmaceutical Manufacturers in Ethiopia, 2024.**

| Items | Status | Number and percentage of rating scores for each company (Companies 1–6) | | | | | |
|---|---|---|---|---|---|---|---|
| | | 1 | 2 | 3 | 4 | 5 | 6 |
| Quality Assurance: Total assessed items #9 | Not compliant | – | – | – | – | – | – |
| | Partially compliant | 5(55) | – | – | – | – | – |
| | Fully compliant | 4(45) | 9(100) | 9(100) | 9(100) | 9(100) | 9(100) |
| QRM system: Total assessed items: #7 | Not compliant | 1(14.3) | – | – | – | – | – |
| | Partially compliant | 6(85.7) | 1(14.3) | 1(14.3) | 1(14.3) | 5(71.4) | 5(71.4) |
| | Fully compliant | | 6(85.7) | 6(85.7) | 6(85.7) | 2(28.6) | 2(28.6) |
| Premise and facility: Total assessed items: #10 | Not compliant | 2(20) | – | – | – | – | – |
| | Partially compliant | 4(40) | 1(10) | 1(10) | 1(10) | 1(10) | 1(10) |
| | Fully compliant | 4(40) | 9(90) | 9(90) | 9(90) | 9(90) | 9(90) |
| Personnel: Total assessed items: #6 | Not compliant | – | – | – | – | – | – |
| | Partially compliant | 2(33.3) | 1(16.7) | 1(16.7) | 1(16.7) | 1(16.7) | 4(66.7) |
| | Fully compliant | 4(66.7) | 5(83.3) | 5(83.3) | 5(83.3) | 5(83.3) | 2(33.3) |
| Hygiene and sanitation: Total assessed items: #9 | Not compliant | – | – | – | – | – | – |
| | Partially compliant | 2(22.2) | – | – | 2(22.2) | – | – |
| | Fully compliant | 7(77.8) | 9(100) | 9(100) | 7(77.8) | 9(100) | 9(100) |
| Validation system: Total assessed items: #8 | Not compliant | – | – | – | – | – | – |
| | Partially compliant | – | – | 1(12.5) | 1(12.5) | – | – |
| | Fully compliant | 8(100) | 8(100) | 7(87.5) | 7(87.5) | 8(100) | 8(100) |
| Documentation Total assessed items: #13 | Not compliant | | – | – | – | – | – |
| | Partially compliant | 2(15.4) | | | 4(30.8) | | |
| | Fully compliant | 11(84.6) | 13(100) | 13(100) | 9(69.2) | 13(100) | 13(100) |
| HVAC system: Total assessed items: #10 | Not compliant | – | – | – | – | – | – |
| | Partially compliant | 3(30) | 1(10) | – | 1(10) | – | – |
| | Fully compliant | 7(70) | 9(90) | 10(100) | 9(90) | 10(100) | 10(100) |
| Equipment: Total assessed items: #8 | Not compliant | – | – | – | | – | – |
| | Partially compliant | 6(75) | 2(25) | 2(25) | 1(12.5) | – | 1(12.5) |
| | Fully compliant | 2(25) | 6(75) | 6(75) | 6(75) | 8(100) | 7(87.5) |
| Materials: Total assessed items: #13 | Not compliant | – | – | – | – | – | – |
| | Partially compliant | 1(7.7) | 2(15.4) | 2(15.4) | – | – | – |
| | Fully compliant | 12(92.3) | 11(84.6) | 11(84.6) | 13(100) | 13(100) | 13(100) |
| Production process: Total assessed items: #8 | Not compliant | – | – | – | – | – | – |
| | Partially compliant | 1(12.5) | 1(12.5) | 1(12.5) | 1(12.5) | 1(12.5) | 1(12.5) |
| | Fully compliant | 7(87.5) | 7(87.5) | 7(87.5) | 7(87.5) | 7(87.5) | 7(87.5) |
| Quality control: Total assessed items: #11 | Not compliant | – | – | 1(9.1) | – | – | – |
| | Partially compliant | 1(9.1) | 1(9.1) | – | – | 1(9.1) | – |
| | Fully compliant | 10(90.9) | 10(90.9) | 10(90.9) | 11(100) | 10(90.9) | 11(100) |
| Regulatory compliance: Total assessed items: #2 | Not compliant | – | – | – | – | 1(50) | – |
| | Partially compliant | 2(100) | – | – | – | – | – |
| | Fully compliant | – | 2(100) | 2(100) | 2(100) | 1(50) | 2(100) |
| Internal audit: Total assessed items: #4 | Not compliant | – | 1(25) | – | 1(25) | 1(25) | – |
| | Partially compliant | 4(100) | 3(75) | 4(100) | 3(75) | 3(75) | 4(100) |
| | Fully compliant | – | – | – | – | – | – |

*(Continued)*

| Items | Status | Number and percentage of rating scores for each company (Companies 1–6) | | | | | |
|---|---|---|---|---|---|---|---|
| | | 1 | 2 | 3 | 4 | 5 | 6 |
| Compliant and recall: Total assessed items: #4 | Not compliant | – | – | 1(25) | – | 1(25) | – |
| | Partially compliant | 1(25) | 1(25) | – | – | 1(25) | 2(50) |
| | Fully compliant | 3(75) | 3(75) | 3(75) | 4(100) | 2(50) | 2(50) |
| Supplier management: Total assessed items: #4 | Not compliant | – | – | – | – | – | – |
| | Partially compliant | 2(50) | – | 4(100) | 2(50) | – | – |
| | Fully compliant | 2(50) | 4(100) | – | 2(50) | 4(100) | 4(100) |

Source: Checklist Survey on GMP Compliance in Ethiopian Pharmaceutical Companies, July 10 to September 16, 2024.

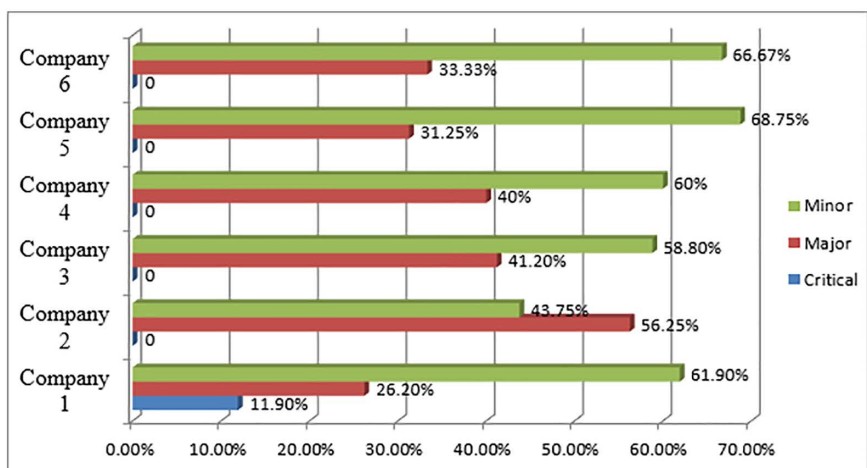

**Fig 1.** Categorization of GMP deficiencies in pharmaceutical manufacturers in Ethiopia, 2024.

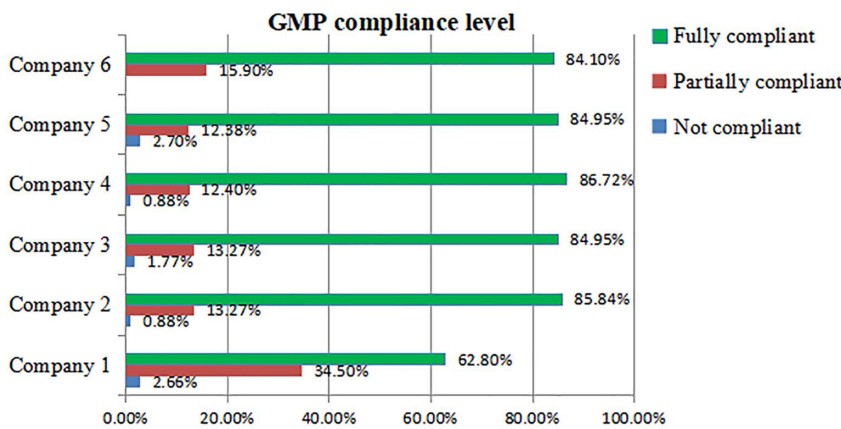

**Fig 2.** GMP compliance level of pharmaceutical manufacturers in Ethiopia, 2024.

   

Compliant," and a low percentage of "Not Compliant." On the other hand, Company 1 has the lowest percentage of "Fully Compliant" (62.8%) and the highest percentage of "Partially Compliant" (34.5%).

Overall, the percentage of "Not Compliant" is generally low across all companies, except for Company 5, where it stands out as higher than the others.

Documentation systems at Companies 1 and 4 were weak, while Companies 1, 2, and 4 had deficiencies in their HVAC systems, particularly in air exchange rates, air filtration, and pressure differentials. Companies 1 and 5 exhibited consistent deficiencies, especially in QA systems (Company 1), QRM, and equipment management (Companies 1 and 2). Both Companies 1 and 5 also showed shortcomings in regulatory compliance. Company 1 had issues with personnel and premises, while Company 5 showed deficiencies in its documentation. Additionally, Companies 2, 4, and 5 had weak internal audit systems. Product complaint and recall systems were poorly implemented at Companies 3 and 5, while Companies 1, 3, and 4 had weak supplier management programs.

The quality assessment has revealed both acceptable and inadequate performance across several areas. The findings on overall GMP compliance, as shown in Fig 3, highlight significant deficiencies in multiple quality elements. Areas identified for improvement include supplier management, complaint and recall systems, internal audits, regulatory compliance, quality control, materials, equipment, HVAC systems, documentation, validation systems, hygiene and sanitation, personnel, and the QRM system.

Additionally, supplier management, complaint and recall systems, internal audits, equipment, premises and facilities, QRM systems, and quality assurance were found to be inadequate. Furthermore, quality control, production processes, equipment, HVAC systems, personnel, and the QRM system were deemed acceptable in some companies.

### Relationship between cGMP implementation status and potential contributing factors

The assessment of various operational variables within the companies revealed notable trends concerning training, workload, government support, production line suitability, GMP certification, and process analytical technology, as outlined in Table 2.

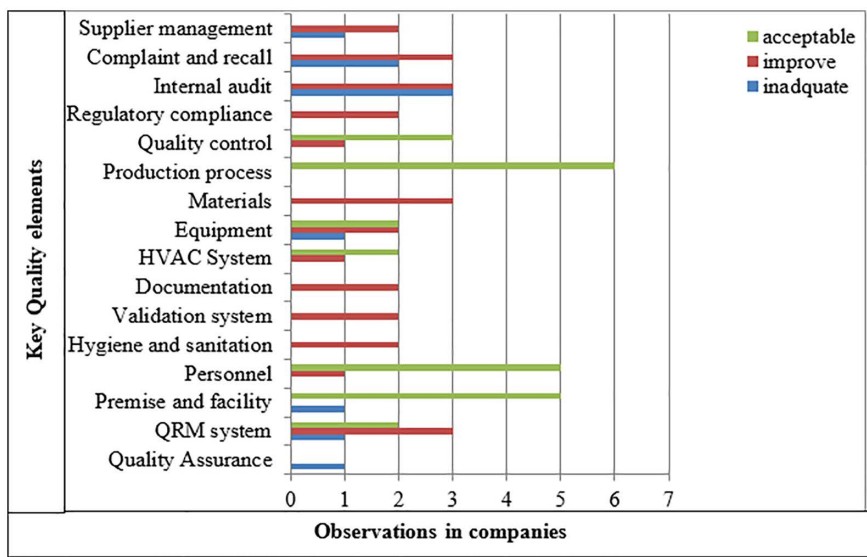

**Fig 3. Overall GMP deficiency categorization of participating companies to key quality elements (N = 6).**

**Table 2. Profile of Pharmaceutical Manufacturing Companies in Ethiopia, 2024 (N = 6).**

| Variables | Status | Frequency | Companies | | | | | |
|---|---|---|---|---|---|---|---|---|
| | | | 1 | 2 | 3 | 4 | 5 | 6 |
| Training | Partially trained | 4 | √ | √ | | | √ | √ |
| | Fully Trained | 2 | | | √ | √ | | |
| Work load | High | 2 | √ | | | | √ | |
| | Medium | 3 | | | √ | √ | | √ |
| | Low | 1 | | √ | | | | |
| Government support | Yes | 4 | √ | √ | √ | √ | | |
| | No | 2 | | | | | √ | √ |
| Production line suitability | Suitable for production process | 4 | | √ | √ | √ | | √ |
| | Not Suitable for production process | 2 | √ | | | | √ | |
| GMP Certificate | Certified | 4 | | √ | √ | √ | | √ |
| | Not certified | 2 | √ | | | | √ | |
| In process Analytical Technology | Somewhat implemented | 1 | | √ | | | | |
| | Not implemented | 5 | √ | | √ | √ | √ | √ |
| GMP compliance | Low compliance | 1 | √ | | | | | |
| | Medium compliance | 3 | | | √ | | √ | √ |
| | High compliance | 2 | | √ | | √ | | |

'√' means 'Yes', 1,2,3,4,5 ang 6 represent companies.

Data was collected through interviews with key personnel from six companies. The findings revealed that two companies reported a high workload, with employees facing substantial demands in terms of task volume, deadlines, and intensity. Employees often manage multiple responsibilities in a fast-paced environment.

Three companies reported a medium workload, where employees experience a balanced workload with manageable tasks and reasonable expectations. One company reported a low workload, focusing on providing employees with lighter, more manageable responsibilities.

Similarly, the interviews conducted on production line suitability revealed that four companies reported having a suitable production line. These companies described their production lines as well-aligned with their products, equipped with efficient tools that are fit for purpose. The premises were adequate, offering sufficient space for production and facilitating a smooth workflow. In contrast, two companies reported that their production lines were unsuitable. The premises were considered inadequate, either due to space limitations or poorly designed layouts that affect production. Additionally, the equipment in these companies was outdated or not well-suited to the demands of modern production. All companies, except one, operate multiple production lines; one company operates a single production line. Regarding government support, four companies reported receiving support in the form of tax-free imports of raw materials, while two companies did not receive such support.

Two companies reported providing partial training. These organizations described their training programs as limited, focusing primarily on specific job-related skills or addressing immediate operational needs. The training is typically shorter in duration and often does not cover broader areas, such as cGMP requirement, QRM system and ongoing quality monitoring.

Four companies reported providing full training. These companies offer comprehensive training programs that cover a wide range of areas, including focusing primarily on specific job-related skills or addressing immediate operational needs.

One company reported partial implementation of PAT, incorporating some automated equipment and limited in-process testing to monitor quality. However, a comprehensive PAT system has not yet been fully implemented. The remaining five

companies have not adopted PAT, relying on manual processes with minimal use of automated equipment and information technology. In these organizations, in-process testing is unstructured, and there are few systems in place to monitor or analyze production data in real-time, hindering operational efficiency.

Finally, the data also revealed that four companies are certified, while two companies are not.

The certified companies hold various industry-recognized certifications, which serve to enhance their credibility and ensure they meet specific quality and regulatory standards.

The association with GMP compliance level and influencers or factors was analyzed by cross-tabulation. A Fisher-Freeman-Halton exact test was performed to determine whether these factors were significantly associated with GMP compliance level with different categories. The results revealed that factors such as workload (which exhibited a negative association, $p = 0.003$) and training ($p = 0.002$) were significantly associated with GMP compliance. However, other factors, such as GMP certification and government support, did not observe a significant relation. The overall association is presented in Table 3.

The results indicated that "Workload" ($p = 0.003$) has a significant negative association with Risk Management Practices. However, other variables such as "Training," "Current Work Experience," and "Government Support" did not show significant associations with RMP.

The demographic characteristics of the participants are depicted in Table 4.

## Discussion

### GMP compliance level of ethiopian pharmaceutical manufacturers per local or WHO standards

This study evaluated GMP compliance levels across six pharmaceutical manufacturers in Ethiopia. The analysis indicated generally high compliance with cGMP, with companies demonstrating adherence rates ranging from 62.8% to 86.72%. Notably, Company 4 exhibited the highest compliance rate at 86.72%, reflecting good adherence to cGMP, which is crucial for maintaining product integrity and ensuring consumer safety [4]. WHO standards state that to guarantee high-quality products, every stage of the manufacturing process must follow particular protocols that have been approved for production and marketing. This is achieved through the application of Good Automated Manufacturing Practice (GAMP) and GMP [13,14]. Automated systems can also lower production costs by minimizing labor requirements and optimizing resource utilization. Additionally, technologies such as real-time monitoring facilitate the early detection of issues, thereby minimizing downtime and waste. Therefore, it is recommended that companies prioritize and implement this strategy [15].

Table 3. Association of GMP Compliance and Risk Management Practices with Factors Fisher-Freeman-Halton Test Results for GMP Compliance Status.

| Variable | Test Statistic | Exact Sig. (2-sided) |
|---|---|---|
| Work load | 12.610 | .003 |
| Training | 11.658 | .002 |
| Certification status | 2.381 | 0.6 |
| Production line | 2.381 | 0.6 |
| Government support | 1.570 | 1.00 |
| Total work experience | 2.204 | 0.487 |
| Current work experience | 5.135 | 0.274 |
| Education level | 2.628 | 0.977 |
| PAT | 2.381 | 0.60 |

PAT = Process Analytical Technology.

**Table 4. Demographic characteristics of Respondents of pharmaceutical manufacturers (N = 30).**

| Respondent's Education levels | Frequency | Percent |
|---|---|---|
| M.Sc. in pharmacy | 5 | 16.7 |
| Bachelor of Pharmacy degree | 19 | 63.3 |
| M.Sc. in chemistry related fields | 2 | 6.7 |
| Bachelor of Science in Biology or Chemistry | 4 | 13.3 |
| **Respondent's department** | | |
| Production | 19 | 63.3 |
| Quality control | 4 | 13.3 |
| Quality Assurance | 6 | 20 |
| Research and development | 1 | 3.3 |
| **Respondents' roles in department** | | |
| General manager | 4 | 13.3 |
| QC manager | 4 | 13.3 |
| Production manager | 1 | 3.3 |
| QA manager | 5 | 16.7 |
| In process QA manager | 1 | 3.3 |
| Syrup and ointment division | 1 | 3.3 |
| Production division head | 3 | 10 |
| Technical manager | 4 | 13.3 |
| Quality Assurance | 6 | 20 |
| Research and development | 1 | 3.3 |
| **Respondent's Total work Experience** | | |
| 6-10 years | 8 | 26.7 |
| >10 years | 22 | 73.3 |
| **Respondent's current position work experience** | | |
| 1-5 years | 2 | 6.7 |
| 6-10 years | 16 | 53.3 |
| >10 years | 12 | 40 |

An overview of the quality elements in pharmaceutical manufacturing reveals several key observations. Company 1 requires improvements in multiple areas, particularly in its QA systems, including controls on starting materials, intermediate products, bulk products, in-process controls, calibrations, and equipment design. Companies 1, 5, and 6 also need to strengthen their QRM processes. Equipment and material management systems at Companies 1, 2, and 3 require attention, while Company 5 should refine its production processes, particularly in process sample testing and the use of dedicated tools for products sensitive to contamination. Since accurate and thorough documentation is critical for compliance and tracking product history, Companies 1 and 4 need to improve their documentation practices. Finally, Companies 4, 5, and 6 must strengthen their product recall procedures, especially in areas such as initiating recalls promptly, assigning responsibilities for complaints and recalls, and investigating non-conformances [4].

A 2022 study in Kabul, Afghanistan, identified product recall (12.98%) and quality assurance (16.44%) as the least compliant domains. In contrast, the current study found that quality assurance was the most compliant area across all companies, except for Company 1, which had a compliance rate of 44.45% [16]. Quality risk management was demonstrated as a weak point across all companies, indicating the need for further improvements. Additionally, employee workload was shown as an issue in four companies: Company 1, Company 3, Company 4, and Company 5. High workloads can directly or indirectly impact GMP compliance; it is crucial to address this issue in order to prevent further

non-compliance. Furthermore, the ownership structure—private or state-owned—may impact the degree of GMP compliance. A study by Hong Chen et al. (2023) found that foreign-owned and private companies generally achieved higher GMP compliance rates compared to state-owned enterprises. In contrast, all companies in the current study were privately owned, with most demonstrating good compliance, except for Company 1, which demonstrated partial compliance [17].

A 2019 study in Kenya by Vugigi et al. evaluated 16 pharmaceutical manufacturers and found that while nine met GMP standards for manufacturing premises, HVAC, water systems, and quality control, they were deficient in areas like research and development (R&D) and the availability of specialized personnel. Similarly, the present study revealed that while all companies adhered to GMP standards for quality control, only Company 1 failed to meet standards for HVAC and quality assurance [18].

Deficiencies in GMP were observed across all six companies, but only Company 1 exhibited severe deficiencies (11.9%), with the other 5 companies maintaining comparatively high compliance. A 2021 study by N. Lubowa et al. in Uganda found that 86% of identified non-conformities were classified as major, with production (30.1%), documentation (24.5%), and quality control (17.6%) being the primary areas of concern. However, quality control, facilities, and equipment were found to be major non-compliant areas in most of organizations in the current study [19].

According to the EFDA's retrospective analysis of 277 foreign GMP inspection reports from 2016 to 2021, 54% of evaluated companies did not adhere to regulatory standards, highlighting the need for corrective action. The most common deficiencies were found in premises and equipment (67%), material management (55%), quality control (53%), and documentation (52%). In contrast, the present study identified only one company with critical deficiencies, with the highest percentages of deviation observed in premises (65%) and equipment (75%) [12].

The analysis also revealed that most companies displayed a high percentage of minor deficiencies, followed by moderate to low major deficiencies. Only one company demonstrated a critical deficiency according to local GMP guidelines. Compliance in the remaining companies could be influenced by the design and establishment of equipment and facilities, which vary depending on the company's year of establishment [8].

Two pharmaceutical companies have fully trained employees, whereas four have partially trained employees. This discrepancy puts the quality and safety of the products at risk because poorly trained individuals can cause problems. A study by R. Wolfle (2021) found that insufficient training and qualifications contribute to 26.3% of quality deviations in GMP-compliant production processes [15].

In conclusion, it is required to work closely with these individual GMP key elements in order to minimize risk, as all of them are directly or indirectly related to GMP compliance, and noncompliance directly affects the quality of the products. This suggests that appropriate corrective actions should be taken promptly; otherwise, it may cause further damage to product quality and compromise patient safety [2,3].

The association between GMP compliance levels and different characteristics inside pharmaceutical firms were investigated using a crosstabulation analysis with the Fisher-Freeman-Halton Test. Notable relationships include workload (p = 0.03), where an increase in workload is associated with a decrease in GMP implementation, and training (p = 0.002), where enhanced training is linked to improved GMP compliance. These findings suggest that strengthening quality assurance and risk management practices could significantly improve overall GMP compliance [20].

However, some variables did not significantly associate with GMP compliance. Interestingly, there was no discernible association between automated technology utilization and GMP compliance level. The absence of association between this variable and GMP compliance levels may be explained by the fact that GMP is a minimum requirement and does not directly influence the adoption of advanced technology or scientific innovations [21]. It is also possible that one company may prioritize certain quality elements, while another focuses on different aspects, leading to variability in compliance. This lack of uniformity likely contributed to the absence of statistically significant associations between these variables. Additionally, government support in the country appears insufficient to influence companies, as the level of support does not create meaningful differences between those that receive assistance and those that do not.

## Conclusions

This study evaluated GMP compliance levels across six pharmaceutical manufacturers in Ethiopia, finding that most companies demonstrated good GMP compliance with adherence rates ranging from 62.8% to 86.72%. However, notable weaknesses were found in areas like quality risk management, documentation, material control, and employee workload. Four companies reported high employee workloads, which may adversely affect GMP compliance. Similarly, four companies had partially trained staff, which could endanger the quality of the product. The report also pointed out technological and resource inadequacies, with only one company utilizing PAT to some extent and only four companies holding GMP certification. The Fisher-Freeman-Halton Test revealed that workload and training were important determinants of compliance, indicating that strengthening these areas will improve overall product safety and GMP adherence.

## Recommendations

Given that many staff in the surveyed companies are only partially trained, investing in regular training programs covering GMP, risk management, and regulatory compliance is essential. Top management should prioritize training and support continuous professional development, with assistance from the EFDA in providing resources. Pharmaceutical companies must also formalize their risk management systems, implementing comprehensive Risk Mitigation Plans with clear, actionable steps and regularly updated risk assessments. Additionally, there is a need for increased investment in advanced technologies like PAT to improve product quality and reduce errors, with support from the Ethiopian government, WHO, and NGOs. Preventive maintenance programs should also be expanded to address equipment failure risks. All companies should obtain GMP certification and strengthen internal regulatory procedures to comply with evolving standards, with regular audits and inspections by the EFDA. To minimize human error and improve GMP compliance, workload management should be addressed, possibly through automation or additional staff.

## Limitations of the study

The purpose of this study was to assess cGMP compliance systems in Ethiopian pharmaceutical manufacturers, serving as a basis for further investigation. However, the study was unable to monitor changes in performance over time because it was cross-sectional. Additionally, the survey was conducted solely among six pharmaceutical manufacturers located in Addis Ababa and Sheger City, with a limited number of key respondents participating. The study lacked representation across the full organizational hierarchy, from security personnel to top-level executives. Therefore, future nationwide surveys and intervention studies are needed to provide a more comprehensive understanding of the cGMP compliance system and identify areas for improvement based on the findings of this study.

## Operational definitions

**Quality Assurance (QA):** involves establishing a dedicated QA department with clearly defined responsibilities. All QA activities are thoroughly documented to ensure traceability and transparency throughout all processes. Furthermore, all production and control operations are specified in writing, providing clear guidelines for consistency and quality in pharmaceutical manufacturing.

**Quality risk management (QRM):** It involves the documentation and regular review of risk assessments, the establishment of clearly defined and accessible QRM procedures, and the implementation of standardized methodologies for risk assessment to ensure consistency and reliability in evaluating risks within pharmaceutical manufacturers.

**Non-compliance with cGMP:** refers to the failure to meet the specified requirements of GMP within the planned arrangements.

**cGMP compliance rating score**: 85% - 100% = high, (70% - 84%) = medium, and <70% = low compliance

**Partially compliance (PC)**: partial execution of the cGMP requirements activity mentioned in the WHO and EFDA cGMP guidelines.

**Full compliance (FC):** cGMP elements and required activities mentioned in WHO and EFDA guidelines are in place, as required.

**Partially trained**: This employee has received partial training in cGMP requirements, including QA, specific training on hazard and contamination, and on-the-job training.

**Fully trained**: This employee has received general training in cGMP requirements, including QA, specific training on hazard and contamination, and on-the-job training.

**Minor deficiency**: An observation describing a situation that is a departure from cGMP but has no significant impact on the product quality.

**Major deficiency**: is an observation characterizing a circumstance that might affect the product, however not to the same extent as a critical observation.

**A critical deficiency** is an observation that indicates a circumstance that is likely to lead to a non-compliant product or that could provide a latent or imminent risk to health.

**Acceptable:** Compliance with GMP for a key quality element is rated "Acceptable" if only minor observations are noted in related areas.

**Improve:** Compliance is rated "Requires improvement" (or "improve") if only few "major" deficiencies (< 5) were observed. on areas related to this specific key quality element.

**Inadequate:** Compliance is deemed "Inadequate" if at least one critical observation or more than five major observations are found in related areas, or if the entire quality element is unavailable at the company.

**Production Line suitability:** It refers to the number of products produced, the appropriateness of the design and installation of equipment, and the suitability of the facility and premises within the company.

**Process Analytical Technology (PAT):** Refers to in-process testing, the availability of automated equipment, and the use of information technology within the company.

**High workload**: Employees are dealing with significant pressures related to task volume, deadlines, and intensity. They frequently juggle various responsibilities in a high-speed work environment.

**Medium workload**: Employees have balanced workload with tasks that are manageable

**Low workload**: refers to Concentrating on assigning employees lighter, more manageable tasks.

## Supporting information

**S1 Table. Supplementary file; General Overview of the GMP Sub-Elements Compliance Scores of Pharmaceutical Manufacturers in Ethiopia, 2024 (n = 6).**
(DOCX)

**S1 Checklist. Supplementary file; The checklist developed for assessment of cGMP in pharmaceutical manufacturers.**
(DOCX)

**S1 File. Supplementary Tables and Assessment Checklist.**
(XLS)

## Acknowledgments

We would like to thank Ethiopian pharmaceutical manufacturers for their collaboration during the data collection process. We are also grateful to Wollega University and Jimma University for their professional support.

## Author contributions

**Conceptualization:** Teka Benti Adola, Dereje Kebebe Borga.

**Data curation:** Teka Benti Adola, Dereje Kebebe Borga.

**Formal analysis:** Teka Benti Adola, Dereje Kebebe Borga.

**Funding acquisition:** Dereje Kebebe Borga.

**Investigation:** Teka Benti Adola, Dereje Kebebe Borga.

**Methodology:** Teka Benti Adola, Desta Assefa, Dereje Kebebe Borga.

**Project administration:** Teka Benti Adola, Dereje Kebebe Borga.

**Resources:** Teka Benti Adola, Dereje Kebebe Borga.

**Software:** Teka Benti Adola, Dereje Kebebe Borga.

**Supervision:** Teka Benti Adola, Dereje Kebebe Borga.

**Validation:** Teka Benti Adola, Dereje Kebebe Borga.

**Visualization:** Teka Benti Adola, Dereje Kebebe Borga.

**Writing – original draft:** Teka Benti Adola, Dereje Kebebe Borga.

**Writing – review & editing:** Teka Benti Adola, Desta Assefa, Fikadu Ejeta, Fanta Gashe, Getahun Paulos, Dereje Kebebe Borga.

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
