## [Decision Letter · Decision Letter 0]

18 Nov 2025

Dear Dr. Adola,

We look forward to receiving your revised manuscript.

Kind regards,

Maher Darwish

Academic Editor

PLOS ONE

Journal Requirements:

Reviewer's Responses to Questions

**Comments to the Author**

1. Is the manuscript technically sound, and do the data support the conclusions?

Reviewer #1: Partly

Reviewer #2: Yes

Reviewer #3: Yes

Reviewer #4: No

2. Has the statistical analysis been performed appropriately and rigorously?

Reviewer #1: I Don't Know

Reviewer #2: Yes

Reviewer #3: Yes

Reviewer #4: Yes

3. Have the authors made all data underlying the findings in their manuscript fully available?

Reviewer #1: No

Reviewer #2: Yes

Reviewer #3: Yes

Reviewer #4: Yes

4. Is the manuscript presented in an intelligible fashion and written in standard English?

Reviewer #1: Yes

Reviewer #2: Yes

Reviewer #3: Yes

Reviewer #4: No

Reviewer #1: This is a cross-sectional, facility-level assessment of cGMP implementation in six large-scale Ethiopian pharmaceutical manufacturers. The authors used a WHO-based checklist (S2) and scored sub-elements (FC=3, PC=2, NC=1) to produce company compliance percentages and identify weak areas (QA, QRM, HVAC, documentation, internal audit, complaint/recall). Strengthening local manufacturing quality systems is high priority worldwide; the paper addresses an important regulatory and public-health issue. Comparable regional analyses exist and make this work useful for national stakeholders.

However, few concerns have been raised for this paper. First, selection of only 6 companies, the sampling strategy risks selection bias, the manuscript does not describe how companies were recruited or whether non-participation occurred (response rate among 10 approached). This affects possible non-response bias, although I understand that companies might not be very cooperative sometimes. Second, The numerical conversion (FC=3, PC=2, NC=1) and the derived % thresholds ( >85%=high, 80–85%=medium, <80%=low) are not justified/validated. No sensitivity analysis or inter-rater reliability reported to show reproducibility between the two pharmacist assessors. Third, only 5 personnel per company (N=30) were interviewed, with such small cell counts the meaningfulness of some p-values is questionable and should be reported with caution. Fourth, no multivariable analysis to control confounding (e.g., company age, number of lines, certification status). Fifth, regarding data availability, authors claim “All relevant data are within the manuscript and Supporting Information” but raw checklists / per-company itemized scores are absent. For reproducibility, provide de-identified item-level datasets or an aggregated CSV in a public repository.

I recommend the authors to explain how the 6 companies were chosen from the 10 approached and discuss representativeness relative to the 12 large manufacturers in Ethiopia in a more detailed manner. Also, I advise the authors to report inter-rater reliability (Cohen’s kappa or % agreement) between the two pharmacists who collected/check-listed data. Provide the full checklist scoring rubric in the supplement (item-level scores per company, de-identified). Please add a methods paragraph justifying FC/PC/NC numeric mapping, explain why thresholds (>85%, 80–85%, <80%) were chosen (cite similar validation if available), or present sensitivity analyses showing how results shift with alternative cutoffs. Discuss selection bias, small sample size, potential Hawthorne effect (sites may temporarily improve practice during audit), lack of PAT adoption measurement detail, and language/translation limitations if any.

With the methodological clarifications and stronger transparency (itemized data, reliability stats) requested above, the manuscript could be acceptable.

Reviewer #2: Review Document

Title of Manuscript:

Assessment of Current Good Manufacturing Practice (cGMP) Compliance in Pharmaceutical Manufacturers in Ethiopia: A Cross-sectional Descriptive Study

General Comments

The topic is relevant and timely, addressing an important issue for pharmaceutical quality systems in developing countries. However, the manuscript requires significant revision to improve clarity, scientific rigor, consistency, and alignment between citations, text, and methodology.

Specific Comments

1. Title

• The title is clear and informative.

2. Background

• The background provides a good overview; however, there is a major issue with the use of references.

o For example, in paragraph 5, you wrote:

"Historically, most Ethiopian pharmaceutical manufacturers have produced drugs that do not meet GMP standards..."

However, the cited reference actually discusses the historical development of pharmaceutical manufacturing in Ethiopia (e.g., establishment of the first company in 1964, post-1991 expansion, and policy adjustments), not noncompliance with GMP or patient harm.

➤ Action: Review and replace or remove mismatched citations throughout the background section.

• The background should also clearly state the existing research gap or problem that motivated this study.

• To assess cGMP compliance, the background must include the criteria or operational definition used to evaluate “degree of compliance.”

• In the objective section, ensure consistency in terminology — the manuscript alternates between “to evaluate” and “to assess.” Choose one and use it consistently throughout.

3. Methods

• The data collection methods should be presented more clearly and concisely. Avoid repetition.

o You mentioned both:

“Data was collected from key technical and managerial personnel directly involved in implementing product quality assurance systems…”

“A standardized checklist based on national and international GMP standards was developed and employed…”

These statements overlap. Combine and clarify.

• The checklist used should have a proper reference (national/international GMP source).

• The inclusion and exclusion criteria need correction. You stated that small-scale pharmaceutical manufacturers were excluded — however, by definition, pharmaceutical industries are generally large-scale operations. Please revise this point for accuracy.

• Clarify what you mean by “high number of product batches” or “wide range of dosage forms.” Provide scientific or numerical justification for these terms.

• The statement that “five key personnel were purposively chosen from each company” requires justification. Why five? Was this number based on pilot testing, organizational structure, or literature precedent?

• Overall, the Methods section should be rewritten to:

o Remove unnecessary repetitions

o Improve logical flow

o Provide scientific rationale for sampling, inclusion/exclusion criteria, and data collection tools.

4. Results

• You reported that “Of the six participating companies, four are GMP certified.”

➤ Clarify whether the two uncertified companies are actually producing pharmaceuticals without certification. This is a serious issue and needs confirmation and contextual explanation.

• The compliance rating scale (“Not Compliant,” “Partially Compliant,” “Fully Compliant”) should be operationalized — specify the criteria or scoring system used to classify each category.

• The statement “Data was collected through interviews with key personnel from six companies” belongs in the Methods section — please remove or paraphrase it in the Results.

• When stating that “Most companies demonstrated a good level of compliance, though deficiencies ranged from minor to critical,” please define what constitutes minor and critical deficiencies.

5. Discussion

• The discussion compares the findings to studies conducted in large-scale industries, which may not be directly comparable to your context of six local companies.

➤ Reframe comparisons using studies with similar scale and methodology for more meaningful interpretation.

• Deepen the discussion by interpreting what the findings imply for regulatory oversight, capacity building, and industry practices in Ethiopia.

6. Conclusion

• The conclusion currently reads like a summary of results.

➤ It should instead highlight the key implications of the findings, potential policy or regulatory recommendations, and suggestions for future research.

7. References

• Ensure uniformity in reference formatting according to PLOS ONE style:

o Use et al. consistently.

o Include DOIs for published articles and URLs for non-published sources.

o Ensure consistency in placement of publication year (either after the author or at the end — not mixed).

o Italicize journal names.

• For example, reference #18 places the year differently — revise for consistency across all entries.

Overall Recommendation

The manuscript addresses an important national issue, but substantial revisions are required before it can be considered for publication. The authors should particularly focus on:

• Correcting mismatched citations in the background,

• Clarifying and justifying methodological choices,

• Defining compliance rating criteria, and

• Enhancing the clarity and consistency of references and conclusions.

Reviewer #3: Review – I can recommend publishing but there are some gaps that need to be addressed.

1. The manuscript's purpose is unclear; clarify why readers need to know about cGMP compliance in Ethiopia.

2. Does cGMP non-compliance affect international medicine exports from Ethiopia? Is there a broader impact?

3. Identify the Ethiopian regulatory authority responsible for enforcing WHO GMP compliance.

4. State the penalties for non-compliance in Ethiopia.

5. Define EFDA (Ethiopian FDA) and include it in the abbreviations section.

Reviewer #4: There are multiple grammatical errors in the text. Abbreviations should not be used for the first time in the abstract. There is no realistic conclusion based on the results obtained. The value of the research lies in providing a report to pharmaceutical manufacturers to implement GMP-based reforms, and the article does not mention a specific purpose for publication.

**Do you want your identity to be public for this peer review?** For information about this choice, including consent withdrawal, please see our Privacy Policy

Reviewer #1: No

Reviewer #2: No

Reviewer #3: No

Reviewer #4: No

---

## [Author Response · Author response to Decision Letter 1]

8 Dec 2025

Assessment of current good manufacturing practice (cGMP) compliance in pharmaceutical manufacturers in Ethiopia

Regarding the reviewer's comments, I have corrected the manuscript as suggested. The following information should provide clarification to the reviewer.

The reference issue was fixed by replacing the original reference with the one that contains the information cited in that paragraph.

This study used a WHO-based checklist because it align with national GMP guideline and scored sub-elements (FC=3, PC=2, NC=1) to produce company compliance percentages and identify weak areas (QA, QRM, HVAC, documentation, internal audit, complaint/recall) and raw checklists are uploaded with the file name 's3’.

The numerical conversion (FC=3, PC=2, NC=1) does not mean that the difference between FC and PC is truly the same as the difference between PC and NC. This numerical assignment serves as a code for NC, PC, and FC to categorize each statement on the checklists into non-compliant, partially compliant, and fully compliant, respectively. The final percentage is calculated based on the total count of statements or checklists categorized as above.

The adjusted CGMP compliance: cGMP compliance rating score: 85% - 100% = high, (70% - 84%)= medium, and <70% = low compliance

The risk associated with deficiencies identified during an inspection was the basis for classifying them consistently into Non-Compliant (NC), Partially Compliant (PC), or Fully Compliant (FC) by the two pharmacist assessors, who used the GMP compliance criteria/guidelines.

The assignment of High, Medium, and Low GMP compliance in the pharmaceutical industry is typically performed through a risk-based scoring system applied during internal or regulatory audits/inspections. It is a key part of Quality Risk Management (QRM) as outlined by international guidelines like ICH Q9

Sample size and selection: The restriction to large-scale, high-volume pharmaceutical manufacturers in this study, due to time and budget constraints, limits the generalizability of the findings. This is further addressed in the Limitations section.

Five key personnel were selected from each company: those who deeply understood GMP and GMP-related activities, and upon whom the quality of the prepared pharmaceutical products relied. Additionally, there was no difference in ideas among those five individuals during the interview for each company.

---

## [Decision Letter · Decision Letter 1]

19 Jan 2026

Dear Dr. Adola,

plosone@plos.org . A letter that responds to each point raised by the academic editor and reviewer(s). You should upload this letter as a separate file labeled 'Response to Reviewers'.A marked-up copy of your manuscript that highlights changes made to the original version. You should upload this as a separate file labeled 'Revised Manuscript with Track Changes'.An unmarked version of your revised paper without tracked changes. You should upload this as a separate file labeled 'Manuscript'.

We look forward to receiving your revised manuscript.

Kind regards,

Maher Darwish

Academic Editor

PLOS One

Journal Requirements:

**Additional Editor Comments:**

1. The manuscript's purpose is unclear; clarify why readers need to know about cGMP compliance in Ethiopia.

2. Does cGMP non-compliance affect international medicine exports from Ethiopia? Is there a broader impact?

3. Identify the Ethiopian regulatory authority responsible for enforcing WHO GMP compliance.

4. State the penalties for non-compliance in Ethiopia.

5. Define EFDA (Ethiopian FDA) and include it in the abbreviations section.</samp>

Reviewers' comments:

**Comments to the Author**

Reviewer #1: All comments have been addressed

Reviewer #3: (No Response)

2. Is the manuscript technically sound, and do the data support the conclusions?

Reviewer #1: Yes

Reviewer #3: Yes

3. Has the statistical analysis been performed appropriately and rigorously?

Reviewer #1: Yes

Reviewer #3: Yes

4. Have the authors made all data underlying the findings in their manuscript fully available?

Reviewer #1: Yes

Reviewer #3: Yes

5. Is the manuscript presented in an intelligible fashion and written in standard English?

Reviewer #1: Yes

Reviewer #3: Yes

Reviewer #1: (No Response)

Reviewer #3: The author has not provided a response to my initial questions. I would request the authors to kindly review the inital comments in detail, and provide a response to each question. I am attaching my questions again.

1. The manuscript's purpose is unclear; clarify why readers need to know about cGMP compliance in Ethiopia.

2. Does cGMP non-compliance affect international medicine exports from Ethiopia? Is there a broader impact?

3. Identify the Ethiopian regulatory authority responsible for enforcing WHO GMP compliance.

4. State the penalties for non-compliance in Ethiopia.

5. Define EFDA (Ethiopian FDA) and include it in the abbreviations section.

**Do you want your identity to be public for this peer review?** For information about this choice, including consent withdrawal, please see our Privacy Policy

Reviewer #1: No

Reviewer #3: No

---

## [Author Response · Author response to Decision Letter 2]

1 Feb 2026

Dear reviewers, our responses to your comments are as follows:

1.The manuscript's purpose is unclear; clarify why readers need to know about cGMP compliance in Ethiopia.

Understanding cGMP compliance in Ethiopia is critical because it serves as the bridge between the country's ambitious public health goals and its industrial economic strategy. As Ethiopia strives to reduce its 80–85% reliance on imported medicines, cGMP serves as the mandatory legal framework to ensure that locally manufactured drugs are safe, effective, and free from contamination. Furthermore, achieving these international standards is the only way for Ethiopian pharmaceutical firms to secure WHO Prequalification, allowing them to compete in global export markets and build a resilient, self-sufficient healthcare supply chain. Dear reviewers, these concepts are found in final paragraph of introduction, the first three lines.

2. Does cGMP non-compliance affect international medicine exports from Ethiopia? Is there a broader impact?

In short, cGMP non-compliance acts as a glass ceiling for Ethiopia’s pharmaceutical industry, making international exports virtually impossible since most global markets and procurement agencies, mandate WHO-level certification. Beyond lost revenue, the broader impact is a deepened dependency on expensive imports that drain foreign currency and a heightened risk to public health, as substandard manufacturing can lead to ineffective treatments or the dangerous rise of antimicrobial resistance. Ultimately, failing to meet these standards keeps the local industry in a cycle of "folk-standard" production, preventing Ethiopia from becoming a regional drug-discovery hub and contributing to a brain drain of its most talented scientific professionals. Dear reviewers, these concepts are found in final paragraph of introduction, 3rd line.

3. Identify the Ethiopian regulatory authority responsible for enforcing WHO GMP compliance.

The Ethiopian Food and Drug Authority (EFDA) is the sole national regulatory body responsible for enforcing WHO Good Manufacturing Practice (GMP) compliance within Ethiopia. Established under Proclamation No. 1112/2019, the EFDA ensures that every pharmaceutical facility—both domestic and international—adheres to stringent quality standards before its products can reach the Ethiopian market. Its enforcement toolkit is comprehensive, ranging from mandatory on-site and remote inspections to the "Market Authorization" process, which legally blocks any medicine not produced in a GMP-compliant factory. By conducting routine audits every few years and verifying "Corrective and Preventive Action" (CAPA) plans, the EFDA acts as the country’s primary gatekeeper for pharmaceutical quality and safety. Dear reviewers, these concepts are found in final paragraph of introduction, 4th line.

4. State the penalties for non-compliance in Ethiopia.

Under Proclamation No. 1112/2019, the Ethiopian Food and Drug Authority (EFDA) enforces a tiered system of penalties designed to protect public health by ensuring pharmaceutical quality. Administrative penalties range from the immediate suspension or revocation of a facility’s GMP certificate and Marketing Authorization (MA) to mandatory product recalls and the physical sealing of non-compliant manufacturing sites. Beyond these regulatory actions, violators face severe criminal and financial penalties, including heavy fines and imprisonment for individuals involved in manufacturing falsified or substandard medicines. These measures ensure that quality standards remain a non-negotiable legal obligation, preventing the circulation of ineffective or dangerous drugs within the country. Dear reviewers, these concepts are found in final paragraph of introduction, 11th line.

5. Define EFDA (Ethiopian FDA) and include it in the abbreviations section

The Ethiopian Food and Drug Authority (EFDA) is the federal regulatory body mandated by Proclamation No. 1263/2021 to protect public health by ensuring the safety, quality, and efficacy of medicines, food, medical devices, and cosmetics. Operating as an autonomous agency under the Ministry of Health, it serves as the nation's primary gatekeeper, overseeing the entire lifecycle of health products—from manufacturing inspections and product registration to post-market surveillance. By enforcing international standards like WHO-GMP and regulating the trade of alcohol and tobacco, the EFDA ensures that all health-related products circulating in the Ethiopian market meet strict scientific and legal requirements to prevent disease and disability. Dear reviewers, these concepts are found in final paragraph of introduction, 6th line.

Updates on references

Following the reviewers' suggestions, we have added new information and references (#10 and #11). As a result, the bibliography has been updated and the reference numbering has changed.

WHO References (#3): I updated the "Technical Report Series" (TRS) numbers. Reference #3 is specifically from TRS 1044, which is the landmark 2022 update for sterile manufacturing.

Thank you for your time.

---

## [Decision Letter · Decision Letter 2]

13 Feb 2026

Assessment of current good manufacturing practice (cGMP) compliance in pharmaceutical manufacturers in Ethiopia: Cross-sectional descriptive study

PONE-D-25-40492R2

Dear Dr. Adola,

We’re pleased to inform you that your manuscript has been judged scientifically suitable for publication and will be formally accepted for publication once it meets all outstanding technical requirements.

Kind regards,

Maher Darwish

Academic Editor

PLOS One

Reviewers' comments:

Reviewer's Responses to Questions

**Comments to the Author**

Reviewer #3: All comments have been addressed

2. Is the manuscript technically sound, and do the data support the conclusions?

Reviewer #3: Yes

3. Has the statistical analysis been performed appropriately and rigorously?

Reviewer #3: Yes

4. Have the authors made all data underlying the findings in their manuscript fully available?

Reviewer #3: Yes

5. Is the manuscript presented in an intelligible fashion and written in standard English?

Reviewer #3: Yes

Reviewer #3: All comments have been answered, I have no further questions. I can now recommend publishing. Thank you.

**Do you want your identity to be public for this peer review?** For information about this choice, including consent withdrawal, please see our Privacy Policy

Reviewer #3: No

---

## [Editor Report · Acceptance letter]

PONE-D-25-40492R2

PLOS One

Dear Dr. Adola,

I'm pleased to inform you that your manuscript has been deemed suitable for publication in PLOS One. Congratulations! Your manuscript is now being handed over to our production team.

Kind regards,

on behalf of

Dr. Maher Darwish

Academic Editor

PLOS One